 **RESEARCH ADVANCE**

# Effective control of SARS-CoV-2 transmission between healthcare workers during a period of diminished community prevalence of COVID-19

Nick K Jones[1,2,3,4†], Lucy Rivett[1,2†], Dominic Sparkes[1,2], Sally Forrest[3,4†], Sushmita Sridhar[3,4,5], Jamie Young[6], Joana Pereira-Dias[3,4], Claire Cormie[3,4], Harmeet Gill[3,4], Nicola Reynolds[7], Michelle Wantoch[8,9], Matthew Routledge[1,2], Ben Warne[1,4], Jack Levy[10], William David Córdova Jiménez[10], Fathima Nisha Begum Samad[10], Chris McNicholas[11], Mark Ferris[12], Jane Gray[13], Michael Gill[13], The CITIID-NIHR COVID-19 BioResource Collaboration, Martin D Curran[2], Stewart Fuller[14], Afzal Chaudhry[15], Ashley Shaw[15], John R Bradley[3,16], Gregory J Hannon[13], Ian G Goodfellow[17], Gordon Dougan[3,4], Kenneth GC Smith[3,4], Paul J Lehner[1,3,4], Giles Wright[12], Nicholas J Matheson[1,4,18‡], Stephen Baker[3,4‡], Michael P Weekes[1,3,4‡*]

[1]Department of Infectious Diseases, Cambridge University NHS Hospitals Foundation Trust, Cambridge, United Kingdom; [2]Clinical Microbiology & Public Health Laboratory, Public Health England, Cambridge, United Kingdom; [3]Department of Medicine, University of Cambridge, Cambridge, United Kingdom; [4]Cambridge Institute of Therapeutic Immunology & Infectious Disease (CITIID), Jeffrey Cheah Biomedical Centre, Cambridge Biomedical Campus, University of Cambridge, Cambridge, United Kingdom; [5]Wellcome Sanger Institute, Hinxton, United Kingdom; [6]Academic Department of Medical Genetics, University of Cambridge, Cambridge, United Kingdom; [7]Wellcome-MRC Cambridge Stem Cell Institute, Jeffrey Cheah Biomedical Centre, Cambridge Biomedical Campus, University of Cambridge, Cambridge, United Kingdom; [8]Wellcome - MRC Cambridge Stem Cell Institute, University of Cambridge, Cambridge, United Kingdom; [9]Department of Haematology, School of Clinical Medicine, University of Cambridge, Cambridge, United Kingdom; [10]Institute for Manufacturing, Department of Engineering, University of Cambridge, Cambridge, United Kingdom; [11]Improvement and Transformation Team, Cambridge University Hospitals NHS Foundation Trust, Cambridge, United Kingdom; [12]Occupational Health and Wellbeing, Cambridge University Hospitals NHS Foundation Trust, Cambridge, United Kingdom; [13]Cancer Research United Kingdom Cambridge Institute, University of Cambridge, Cambridge, United Kingdom; [14]National Institutes for Health Research Cambridge Biomedical Research Centre, Cambridge, United Kingdom; [15]Cambridge University Hospitals NHS Foundation Trust, Cambridge, United Kingdom; [16]National Institutes for Health Research Cambridge, Clinical Research Facility, Cambridge, United Kingdom; [17]Division of Virology, Department of Pathology, University of Cambridge, Cambridge, United Kingdom; [18]NHS Blood and Transplant, Cambridge, United Kingdom

**\*For correspondence:**
mpw1001@cam.ac.uk

[†]These authors contributed equally to this work
[‡]These authors also contributed equally to this work

**Group author details:**
The CITIID-NIHR COVID-19 BioResource Collaboration See page 7

**Abstract** Previously, we showed that 3% (31/1032)of asymptomatic healthcare workers (HCWs) from a large teaching hospital in Cambridge, UK, tested positive for SARS-CoV-2 in April 2020. About 15% (26/169) HCWs with symptoms of coronavirus disease 2019 (COVID-19) also tested positive for SARS-CoV-2 (Rivett et al., 2020). Here, we show that the proportion of both asymptomatic and symptomatic HCWs testing positive for SARS-CoV-2 rapidly declined to near-zero between 25th April and 24th May 2020, corresponding to a decline in patient admissions with COVID-19 during the ongoing UK 'lockdown'. These data demonstrate how infection prevention and control measures including staff testing may help prevent hospitals from becoming independent 'hubs' of SARS-CoV-2 transmission, and illustrate how, with appropriate precautions, organizations in other sectors may be able to resume on-site work safely.

## Introduction

The role of nosocomial transmission of SARS-CoV-2 has been highlighted by recent evidence suggesting that 20% of SARS-CoV-2 infections among patients in UK hospitals and up to 89% of infections among HCWs may have originated in hospitals (*Evans et al., 2020*; *Iacobucci, 2020*). Since the introduction of 'lockdown' in the UK, community transmission rates of SARS-CoV-2 have generally declined (*Public Health England (PHE), 2020*). Conversely, concerns have been raised that hospitals could become independent 'hubs' for ongoing SARS-CoV-2 transmission between patients and HCWs, which would effectively prolong the epidemic (*Iacobucci, 2020*). In this context, the evolution of the epidemic curves of a hospital's symptomatic and asymptomatic workforce has not been well described.

We recently initiated a comprehensive HCW screening programme for SARS-CoV-2 in a large teaching hospital in Cambridge, UK. Over a 3-week period from 6th to 24th April 2020, 3% (31/1032) HCWs in the *asymptomatic screening arm*, 15.4% (26/169) HCWs in the *symptomatic screening arm*, and 7.7% (4/52) contacts in the *symptomatic household contact screening arm* tested positive for SARS-CoV-2 (*Rivett et al., 2020*). Our data from the asymptomatic screening arm were consistent with the results of Shields et al. (*Shields et al., 2020*). Over the next 4 weeks from 25th April to 24th May 2020, we performed a further 3388 additional tests. Here, we present these longitudinal data, in the context of the hospital patient population and wider local community.

## Results

Testing for SARS-CoV-2 RNA was performed with real-time RT-PCR using throat and nose swab samples of HCWs from Cambridge University Hospitals NHS Foundation Trust (CUHNFT) and their symptomatic household contacts. Over the new study period (25th April to 24th May 2020), 2611 additional tests were performed in the *HCW asymptomatic screening arm*, 555 additional tests in the *HCW symptomatic screening arm*, and 216 additional tests in the *HCW household contact screening arm*. A further six tests did not have a clearly recorded arm of origin. Over the entire study period, the median age of HCWs and their household contacts was 36.5 and 35.5 years, respectively. About 68.4% were female and 31.6% were male. Of the individuals testing positive over the entire study period, the median age of HCWs and their household contacts was 32 and 47 years, respectively. About 77.9% of all positive tests were from females and 22.1% from males. *Table 1* summarizes the total number of HCWs testing positive through either arm of the screening programme, according to the job role. A comparison of the proportions of hospital employees from each job role that tested positive through the *HCW symptomatic screening arm* revealed no statistically significant difference (Pearson's chi-square test p=0.419). Reasonable comparison of the proportions testing positive through the *HCW asymptomatic screening arm* was not possible due to non-random sampling of different areas of the hospital, meaning some job roles had been more frequently targeted for asymptomatic screening than others.

Between 25th April and 24th May 2020, a total of 34 new positive tests were reported. In the *HCW symptomatic* and *HCW symptomatic household contact screening arms* combined (reflecting all individuals with self-reported symptoms at the time of testing), 13/771 (1.7%) tests were positive,

**Table 1.** Combined data for SARS-CoV-2 RNA positive HCWs by role and screening arm, from the present study and our previous study (**Rivett et al., 2020**).

Difference in proportions of HCWs testing positive through the symptomatic screening arm was analysed using Pearson's chi-square test.

| Role | HCW asymptomatic screening arm | HCW symptomatic screening arm | Total number of hospital employees |
|---|---|---|---|
| Nurse | 25 | 19 | 3621 |
| Healthcare assistant | 14 | 8 | 1734 |
| Doctor | 8 | 6 | 1871 |
| Cleaners | 2 | 3 | 560 |
| Radiographer | 2 | 1 | 217 |
| Radiology support worker | 0 | 1 | 35 |
| Physiotherapist | 1 | 0 | 116 |

Overall, 360 individuals underwent repeat testing, either as part of the asymptomatic screening programme, or for other reasons as previously described (**Rivett et al., 2020**). The median turnaround time from sample arrival in the laboratory to final verification was 18 hr 45 min. Positive results were called out on the same day, with negative results emailed within 24 hr.

which was significantly lower than 30/221 (13%) in the original study period (Fisher's exact test p<0.0001). In the *HCW asymptomatic screening arm,* 21/2611 (0.8%) tests were positive, which again was significantly lower than 31/1032 (3%) in the original study period (Fisher's exact test p<0.0001). As we previously observed (**Rivett et al., 2020**), individuals captured in the *HCW asymptomatic screening arm* were generally asymptomatic at the time of screening; however, these individuals could be divided into subgroups. In the first subgroup, 8/21 (38%) HCWs had no symptoms at all. Of these, 5/8 (63%) remained entirely asymptomatic 5–7 weeks after their positive test, whereas 2/8 (25%) developed symptoms 24–48 hr after testing. One HCW could not be contacted to obtain further history. In the second subgroup, 6/21 (29%) had retrospectively experienced some symptoms prior to screening. Of these, 5/6 (83%) had symptoms with a high pre-test probability of COVID-19 (**Rivett et al., 2020**) commencing >7 days prior to screening, of whom 3/5 had appropriately self-isolated then returned to work, and 1/5 was tested shortly after developing symptoms. 1/6 (17%) had symptoms with a low pre-test probability of COVID-19 (**Rivett et al., 2020**) commencing <7 days prior to screening and had not self-isolated. In the third subgroup, 7/21 (33%) were detected through repeat sampling of HCW who previously tested positive. Of these, 4/7 (57%) were tested to determine their suitability to return to work with severely immunocompromised/immunosuppressed patients, as dictated by UK national guidance (**National Institute for Health and Care Excellence (NICE), 2020**). The remaining 3/7 (43%) were from HCWs tested incidentally for the second time in the asymptomatic HCW screening programme. The median interval between serial positive tests was 16.5 days (IQR 9.5–19.5). All cases were attributable to prolonged SARS-CoV-2 RNA detection from a single infection, rather than re-infection. Our approach to patients with repeatedly positive SARS-CoV-2 PCR tests is described in the Methods.

The fraction of positive tests among the *HCW asymptomatic, and HCW symptomatic* and *household contact screening groups* combined varied over time (**Figure 1A**, **Table 2**). In particular, during the last 2 weeks of the study period (11th to 24th May 2020), we identified only four positive SARS-CoV-2 samples from 2016 tests performed, two from the *HCW asymptomatic* and two from the *HCW symptomatic/symptomatic household contact arms*. This fall in positive HCW tests mirrored the decline in both patients testing positive at CUHNFT and those tested throughout the wider region (**Figure 1B**). Similar trends were observed in a smaller cohort study of HCWs in London (**Treibel et al., 2020**).

In our original study between 6th and 24th April 2020, we described in detail two clusters of HCW infections (**Rivett et al., 2020**). From 25th April to 24th May 2020, we detected one additional cluster on a general medical ward with a separate area for patients with proven COVID-19 and another area for those without. This was identified through targeted screening of the ward over a 24 hr period from 4th to 5th May 2020, in response to four staff testing positive through the *HCW symptomatic arm* of the screening programme from 27th to 30th April 2020. Reactive screening of a

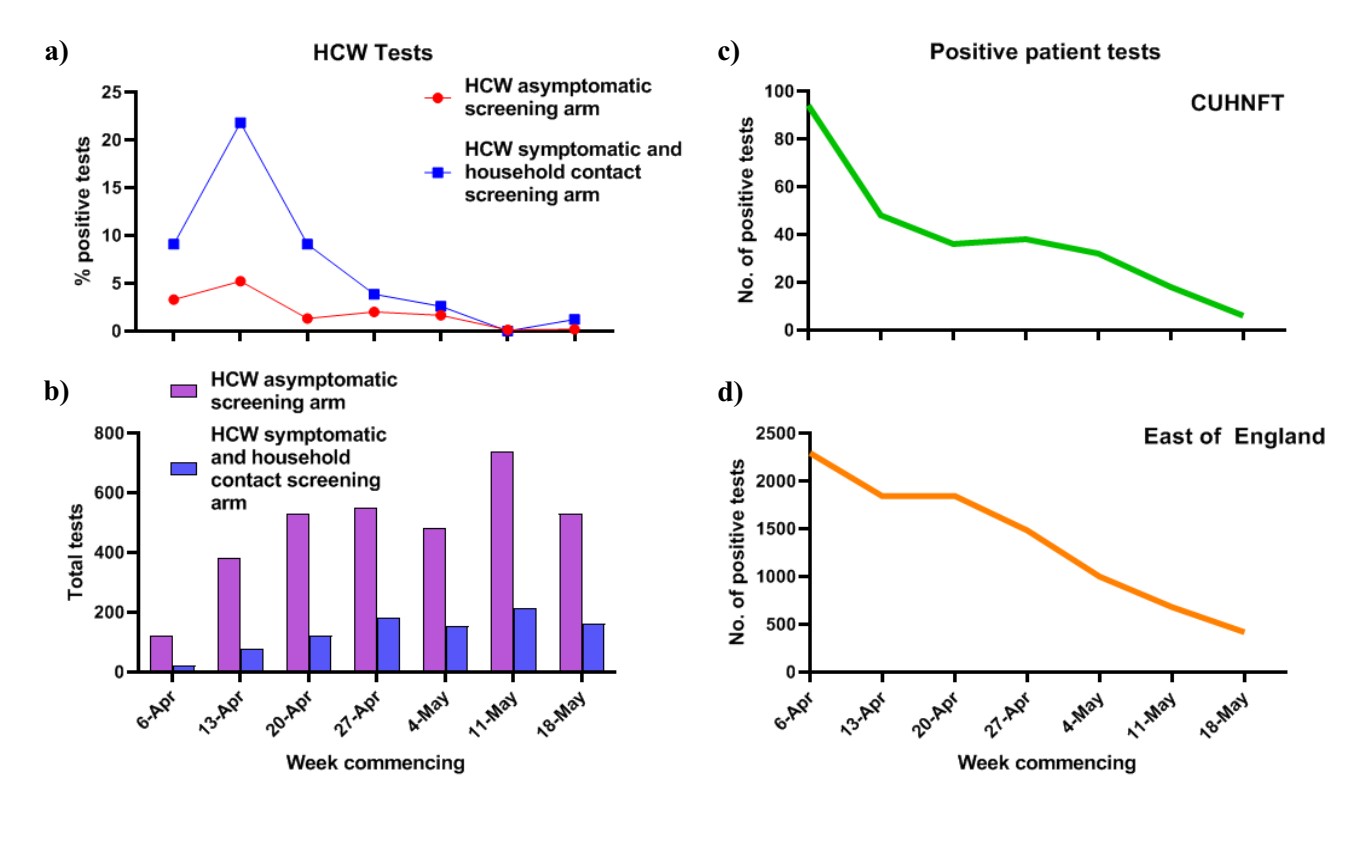

**Figure 1.** Trends in positive SARS-CoV-2 PCR tests among HCWs, hospital patients and the wider community over time. (**a**) Positive SARS-CoV-2 tests for asymptomatic and symptomatic screening arms by week. (**b**) Total HCW SARS-CoV-2 tests in CUHNFT performed by week. (**c**) Total positive SARS-CoV-2 patient tests in Cambridge University Hospital NHS Foundation Trust (CUHNFT) by week. (**d**) Total positive SARS-CoV-2 tests in the East of England (EOE) by week.

The online version of this article includes the following source data for figure 1:

**Source data 1.** Source data for trends in SARS-CoV-2 PCR positive HCWs, hospital patients and individuals in the wider community.

further 40 staff from the same ward identified a further three positive asymptomatic HCWs. In addition, a further two HCWs tested positive in an asymptomatic screen of 30 individuals from a closely related clinical area (designated for non-COVID patients) on 6th May 2020.

## Discussion

Our data demonstrate a dramatic fall in the prevalence of symptomatic and asymptomatic SARS-CoV-2 infection among HCWs in our hospital during the study period. On average, the number of secondary infections among HCWs arising from each infected HCW (effectively, the reproduction number (R) for SARS-CoV-2 transmission between HCWs) must therefore be <1.

As well as an acquisition from other HCWs, infections among HCWs may also be acquired from patients, as well as other individuals outside the hospital. Our study period coincided with a decline in the rate of infection across our local community, and our data are consistent with a reduction in transmission within the hospital, a reduction in community-based acquisition of infection by HCWs, or (most likely) a combination of both. In the absence of detailed epidemiological data, it is not possible to formally differentiate between these possibilities or determine their relative effect sizes. Nonetheless, our identification of HCW infection clusters in specific areas of the hospital highlighted the potential for workplace acquisition of SARS-CoV-2, which may lead to self-sustaining outbreaks if left uninterrupted (*Rivett et al., 2020*; *Meredeth et al., 2020*). For each of these clusters, timely

**Table 2.** Positive tests and total number of SARS-CoV-2 tests performed in each screening arm categorised according to week since starting the healthcare worker testing programme (6th April–24th May 2020).

| | Week commencing | | | | | | | |
| | 6th April | 13th April | 20th April | 27th April | 4th May | 11th May | 18th May | Total |
|---|---|---|---|---|---|---|---|---|
| HCW asymptomatic screening arm | 4/121 | 20/383 | 7/529 | 11/550 | 8/483 | 1/738 | 1/840 | 52/3644 (1.4%) |
| HCW symptomatic screening arm | 1/15 | 14/60 | 11/95 | 7/119 | 3/104 | 0/164 | 2/168 | 38/725 (5.2%) |
| HCW symptomatic household contacts | 1/7 | 3/18 | 0/26 | 0/62 | 1/50 | 0/51 | 0/53 | 5/267 (1.8%) |
| *Unknown* | 0/0 | 0/2 | 0/13 | 0/0 | 0/4 | 0/1 | 0/1 | 0/21 |
| *All* | 6/143 (4.1%) | 37/463 (7.9%) | 18/663 (2.7%) | 18/731 (2.4%) | 12/641 (1.8%) | 1/954 (0.1%) | 3/1062 (0.2%) | 95/4657 (2%) |

identification of HCW infection proved effective in terminating chains of hospital transmission between staff, preventing ongoing nosocomial infection.

With the incidence of infection having fallen significantly in hospitalised patients, HCWs and the wider community, many hospitals across the UK and further afield have been afforded precious time to build the infrastructure necessary to establish comprehensive screening programmes in anticipation of a possible second epidemic peak. For hospitals already operating newly established screening programmes, the challenge now is to up-scale to the point that screening can occur at a frequency that permits pre-symptomatic capture of as close to 100% of all new infections as possible. This approach will enable staff to be removed from the workplace at the time of peak infectivity (*He et al., 2020*). The minimum screening frequency required needs to be carefully modelled, with recent estimates suggesting the need for weekly testing to prevent 16–33% of onward transmission from HCWs, depending on the time taken for results to be reported, and another study estimating the need for daily screening to prevent 65% of HCW-to-HCW transmission events (*Evans et al., 2020*; *Grassly et al., 2020*). In practice, we have observed good results in our hospital with a current frequency of asymptomatic screening every 2–4 weeks. Those being screened are prioritised by anticipated ward-based exposure to COVID-19, with additional targeted screens triggered by excess staff sickness or the identification of symptomatic cases on specific wards (*Rivett et al., 2020*). In addition to asymptomatic screening, testing of symptomatic HCWs is essential for preventing excessive erosion of the hospital workforce by self-isolation on the basis of symptoms alone, and testing of symptomatic HCW household contacts negates the need for unnecessary self-quarantine periods for co-habiting HCWs. We found uptake to the HCW symptomatic household contact screening arm of our programme to be notably lower than the HCW symptomatic arm despite regular communications to advertise the service within CUHNFT. This lack of uptake may reflect a lack of awareness that symptomatic non-HCWs were eligible for testing, provided they shared a household with a hospital employee. Many non-hospital employees may also have been more inclined to attend national testing centres or be less aware of the spectrum of COVID-19 symptoms.

Importantly, our data demonstrate that CUHNFT was not acting as an independent 'hub' for ongoing COVID-19 transmission among HCWs. The absence of nosocomial transmission likely reflects the combined efficacy of HCW testing, stringent prospective, and reactive infection prevention and control measures, and appropriate social distancing among the workforce. These findings should give reassurance to both hospital staff and patients that healthcare facilities remain safe places to give and receive care. Furthermore, since CUHNFT, with approximately 11,000 staff members (many of whom are based in the hospital) is a major regional employer, we predict that comparable

organisations in other sectors may also be able to resume on-site work safely by instigating similar precautions.

## Materials and methods

### Staff screening protocols

We previously described protocols for staff screening, sample collection, laboratory processing, and results reporting in detail (*Rivett et al., 2020*). These methods remained unchanged throughout this study period. Two parallel streams of entry into the testing programme included (i) *HCW symptomatic, and HCW symptomatic household contact screening arms* and (ii) an *HCW asymptomatic screening arm*. In the former, any patient-facing or non-patient-facing HCW could voluntarily refer themselves or a household contact, should they develop symptoms suggestive of COVID-19. In the latter, HCWs could volunteer to take part in a rolling programme of testing for all patient-facing and non-patient-facing staff working in defined clinical areas thought to be at risk of SARS-CoV-2 transmission. Testing was performed (i) at temporary on-site 'Pods'; (ii) via self-swabbing kits delivered to HCWs in their area of work. All individuals in each arm of the programme performed a self-swab at the back of the throat then the nasal cavity, followed by RNA extraction and amplification using real-time RT-PCR (*Sridhar et al., 2020*). Cluster investigation was initiated when three or more HCWs working in the same clinical area tested positive for SARS-CoV-2 in 1 week .

### Management of HCW with repeat positive tests

Current National Institute for Health and Care Excellence (NICE) guidelines require a negative test before returning to work with immunocompromised patients (*National Institute for Health and Care Excellence (NICE), 2020*). In accordance with the UK national guidance, individuals with repeat positive screens following a minimum period of 7 days self-isolation were advised to continue working if they were not scheduled to come into close contact with heavily immunocompromised patients, provided they remained asymptomatic (*UK Government, 2020*). This approach to managing repeat positive screens is further supported by recent data from the Korea Centers for Disease Control and Prevention, which showed no clear evidence of onward transmission to the contacts of 285 repeat-positive individuals, 108 of whom had samples taken for attempted viral culture, which was universally unsuccessful (*Korea Centers for Disease Control & Prevention (KCDC), 2020*). Additional small studies have also demonstrated an inability to culture virus from clinical samples obtained later than 8 days after symptom onset, suggesting prolonged detection of viral RNA is unlikely to indicate an ongoing risk of transmission (*Wölfel et al., 2020*; *Bullard et al., 2020*).

### Data extraction and analysis

Swab result data for HCWs and patients were extracted directly from the hospital-laboratory interface software, Epic (Verona, WI) and from SARS-CoV-2 point of care testing. Data for SARS-CoV-2 infections from the local community were extracted from Public Health England's Data Dashboard (*Public Health England (PHE), 2020*). Data were collated using Microsoft Excel, and figures produced with GraphPad Prism (GraphPad Software, La Jolla, CA). Fisher's exact test was used to compare the proportion of HCWs testing positive in this study period to that of our previous study period (*Rivett et al., 2020*). Pearson's chi-square test was used for comparison of the proportions of HCWs testing positive in each job role.

## Acknowledgements

This work was supported by the Wellcome Trust Senior Research Fellowships 108070/Z/15/Z to MPW, 215515/Z/19/Z to SGB and 207498/Z/17/Z to IGG; Collaborative award 206298/B/17/Z to IGG; Principal Research Fellowship 210688/Z/18/Z to PJL; Investigator Award 200871/Z/16/Z to KGCS; Addenbrooke's Charitable Trust (to MPW, SGB, and PJL); the Medical Research Council (CSF MR/P008801/1 to NJM); NHS Blood and Transfusion (WPA15-02 to NJM); National Institute for Health Research (Cambridge Biomedical Research Centre at CUHNFT), to JRB, AC and GD, Cancer Research UK (PRECISION Grand Challenge C38317/A24043 award to JY).
    **The CITIID-NIHR COVID-19 BioResource Collaboration**

**Principal Investigators**: Stephen Baker, John Bradley, Gordon Dougan, Ian Goodfellow, Ravi Gupta, Paul J Lehner, Paul A Lyons, Nicholas J Matheson, Kenneth GC Smith, M Estee Torok, Mark Toshner, Michael P Weekes

**Infectious Diseases Department:** Nicholas K Jones, Lucy Rivett, Matthew Routledge, Dominic Sparkes, Ben Warne

**SARS-CoV-2 testing team:** Claire Cormie, Sally Forrest, Harmeet Gill, Iain Kean, Joana Pereira-Dias, Nicola Reynolds, Sushmita Sridhar, Michelle Wantoch, Jamie Young

**COG-UK Cambridge Sequencing Team:** Sarah Caddy, Laura Caller, Theresa Feltwell, Grant Hall, William Hamilton, Myra Hosmillo, Charlotte Houldcroft, Aminu Jahun, Fahad Khokhar, Luke Meredith, Anna Yakovleva

**NIHR BioResource:** Helen Butcher, Daniela Caputo, Debra Clapham-Riley, Helen Dolling, Anita Furlong, Barbara Graves, Emma Le Gresley, Nathalie Kingston, Sofia Papadia, Hannah Stark, Kathleen E Stirrups, Jennifer Webster

**Research nurses:** Joanna Calder, Julie Harris, Sarah Hewitt, Jane Kennet, Anne Meadows, Rebecca Rastall, Criona O,Brien, Jo Price, Cherry Publico, Jane Rowlands, Valentina Ruffolo, Hugo Tordesillas

**CRUK:** Michael Gill, Jane Gray, Greg Hannon

**NIHR Cambridge Clinical Research Facility:** Karen Brookes, Laura Canna, Isabel Cruz, Katie Dempsey, Anne Elmer, Naidine Escoffery, Stewart Fuller, Heather Jones, Carla Ribeiro, Caroline Saunders, Angela Wright

**Cambridge Cancer Trial Centre:** Rutendo Nyagumbo, Anne Roberts

**Clinical Research Network Eastern:** Ashlea Bucke, Simone Hargreaves, Danielle Johnson, Aileen Narcorda, Debbie Read, Christian Sparke, Lucy Worboys

**Administrative staff, CUHNFT:** Kirsty Lagadu, Lenette Mactavous

**CUHNFT NHS Foundation Trus**t: Tim Gould, Tim Raine, Ashley Shaw

**Cambridge Cancer Trials Centre:** Claire Mather, Nicola Ramenatte, Anne-Laure Vallier

**Legal/Ethics:** Mary Kasanicki

**CUHNFT Improvement and Transformation Team:** Penelope-Jane Eames, Chris McNicholas, Lisa Thake

**Clinical Microbiology & Public Health Laboratory (PHE):** Neil Bartholomew, Nick Brown, Martin Curran, Surendra Parmar, Hongyi Zhang

**Occupational Health:** Ailsa Bowring, Mark Ferris, Geraldine Martell, Natalie Quinnell, Giles Wright, Jo Wright

**Health and Safety:** Helen Murphy

**Department of Medicine Sample Logistics:** Benjamin J Dunmore, Ekaterina Legchenko, Stefan Gräf, Christopher Huang, Josh Hodgson, Kelvin Hunter, Jennifer Martin, Federica Mescia, Ciara O'Donnell, Linda Pointon, Joy Shih, Rachel Sutcliffe, Tobias Tilly, Zhen Tong, Carmen Treacy, Jennifer Wood

**Department of Medicine Sample Processing and Acquisition:** Laura Bergamaschi, Ariana Betancourt, Georgie Bowyer, Aloka De Sa, Maddie Epping, Andrew Hinch, Oisin Huhn, Isobel Jarvis, Daniel Lewis, Joe Marsden, Simon McCallum, Francescsa Nice, Ommar Omarjee, Marianne Perera, Nika Romashova, Mateusz Strezlecki, Natalia Savoinykh Yarkoni, Lori Turner

**Epic team/other computing support:** Barrie Bailey, Afzal Chaudhry, Rachel Doughton, Chris Workman

**Statistics/modelling**: Caroline Trotter

**Department of Engineering:** William David Cordova Jiménez, Jack Levy, Fatima NB Samad

# Additional information

## Group author details

**The CITIID-NIHR COVID-19 BioResource Collaboration**
Stephen Baker; John Bradley; Gordon Dougan; Ian Goodfellow; Ravi Gupta; Paul J Lehner; Paul A Lyons; Nicholas J Matheson; Kenneth GC Smith; M Estee Torok; Mark Toshner; Michael P Weekes; Nicholas K Jones; Lucy Rivett; Matthew Routledge; Dominic Sparkes; Ben Warne; Claire

Cormie; Sally Forrest; Harmeet Gill; Iain Kean; Joana Pereira-Dias; Nicola Reynolds; Sushmita Sridhar; Michelle Wantoch; Jamie Young; Sarah Caddy; Laura Caller; Theresa Feltwell; Grant Hall; William Hamilton; Myra Hosmillo; Charlotte Houldcroft; Aminu Jahun; Fahad Khokhar; Luke Meredith; Anna Yakovleva; Helen Butcher; Daniela Caputo; Debra Clapham-Riley; Helen Dolling; Anita Furlong; Barbara Graves; Emma Le Gresley; Nathalie Kingston; Sofia Papadia; Hannah Stark; Kathleen E Stirrups; Jennifer Webster; Joanna Calder; Julie Harris; Sarah Hewitt; Jane Kennet; Anne Meadows; Rebecca Rastall; Criona O Brien; Jo Price; Cherry Publico; Jane Rowlands; Valentina Ruffolo; Hugo Tordesillas; Michael Gill; Jane Gray; Greg Hannon; Karen Brookes; Laura Canna; Isabel Cruz; Katie Dempsey; Anne Elmer; Naidine Escoffery; Stewart Fuller; Heather Jones; Carla Ribeiro; Caroline Saunders; Angela Wright ; Rutendo Nyagumbo; Anne Roberts; Ashlea Bucke; Simone Hargreaves; Danielle Johnson; Aileen Narcorda; Debbie Read; Christian Sparke; Lucy Worboys; Kirsty Lagadu; Lenette Mactavous; Tim Gould; Tim Raine; Ashley Shaw; Claire Mather; Nicola Ramenatte; Anne-Laure Vallier; Mary Kasanicki; Penelope-Jane Eames; Chris McNicholas; Lisa Thake; Neil Bartholomew; Nick Brown; Martin Curran; Surendra Parmar; Hongyi Zhang; Ailsa Bowring; Mark Ferris; Geraldine Martell; Natalie Quinnell; Giles Wright; Jo Wright; Helen Murphy; Benjamin J Dunmore; Ekaterina Legchenko; Stefan Gräf; Christopher Huang; Josh Hodgson; Kelvin Hunter; Jennifer Martin; Federica Mescia; Ciara ODonnell; Linda Pointon; Joy Shih; Rachel Sutcliffe; Tobias Tilly; Zhen Tong; Carmen Treacy; Jennifer Wood; Laura Bergamaschi; Ariana Betancourt; Georgie Bowyer; Aloka De Sa; Maddie Epping; Andrew Hinch; Oisin Huhn; Isobel Jarvis; Daniel Lewis; Joe Marsden; Simon McCallum; Francescsa Nice; Ommar Omarjee; Marianne Perera; Nika Romashova; Mateusz Strezlecki; Natalia Savoinykh Yarkoni; Lori Turner; Barrie Bailey; Afzal Chaudhry; Rachel Doughton; Chris Workman; Caroline Trotter; William David; Cordova Jiménez; Jack Levy; Fatima NB Samad

## Competing interests

Afzal Chaudhry: Afzal Chaudhry reports grants from Cambridge Biomedical Research Centre at CUHNFT, during the conduct of the study. Ian G Goodfellow: Ian Goodfellow reports grants from Wellcome Trust (Senior Research Fellowships), grants from Wellcome Trust (Collaborative Award), grants from Addenbrooke's Charitable Trust, during the conduct of the study. Gordon Dougan: Gordon Dougan reports grants from NIHR, during the conduct of the study. Kenneth GC Smith: Kenneth GC Smith reports grants from Wellcome Trust, during the conduct of the study. Paul J Lehner: Paul J Lehner reports grants from Wellcome Trust Principal Research Fellowship, grants from Addenbrooke's Charitable Trust, during the conduct of the study. Nicholas J Matheson: Nicholas J Matheson reports grants from Medical Research Council (Clinician Scientist Fellowship), grants from NHS Blood and Transfusion, during the conduct of the study. Stephen Baker: Stephen Baker reports grants from Wellcome Trust (Senior Research Fellowships), from Addenbrooke's Charitable Trust, during the conduct of the study. Michael P Weekes: Michael P Weekes reports grants from Wellcome Trust (Senior Research Fellowships), from Addenbrooke's Charitable Trust, during the conduct of the study. The other authors declare that no competing interests exist.

## Funding

| Funder | Grant reference number | Author |
|---|---|---|
| Wellcome | 108070/Z/15/Z | Michael P Weekes |
| Wellcome | 215515/Z/19/Z | Stephen Baker |
| Wellcome | 207498?Z/17/Z | Ian G Goodfellow |
| Wellcome | 206298/B/17/Z | Ian G Goodfellow |
| Wellcome | 210688/Z/18/Z | Paul J Lehner |
| Wellcome | 200871/Z/16/Z | Kenneth GC Smith |
| Medical Research Council | MR/P008801/1 | Nicholas J Matheson |
| Addenbrooke's Charitable Trust, Cambridge University Hospitals | | Ian G Goodfellow Paul J Lehner Stephen Baker Michael P Weekes |

| NHS Blood and Transplant | WPA15-02 | Nicholas J Matheson |
| National Institute for Health Research | | Afzal Chaudhry John R Bradley Gordon Dougan |
| Cancer Research UK | C38317/A24043 | Jamie Young |

The funders had no role in study design, data collection and interpretation, or the decision to submit the work for publication.

## Author contributions

Nick K Jones, Lucy Rivett, Dominic Sparkes, Conceptualization, Data curation, Formal analysis, Investigation, Methodology, Writing - original draft, Project administration, Writing - review and editing; Sally Forrest, Jamie Young, Jack Levy, William David Córdova Jiménez, Fathima Nisha Begum Samad, Data curation, Investigation, Methodology, Project administration, Writing - review and editing; Sushmita Sridhar, Joana Pereira-Dias, Claire Cormie, Nicola Reynolds, Michelle Wantoch, Jane Gray, Michael Gill, Ian G Goodfellow, Data curation, Investigation, Methodology, Writing - review and editing; Harmeet Gill, Ashley Shaw, Resources, Data curation, Investigation, Methodology, Writing - review and editing; Matthew Routledge, Resources, Data curation, Methodology, Project administration, Writing - review and editing; Ben Warne, Data curation, Methodology, Writing - review and editing; Chris McNicholas, Data curation, Software, Formal analysis, Writing - review and editing; Mark Ferris, Resources, Investigation, Methodology, Project administration, Writing - review and editing; Martin D Curran, Stewart Fuller, Afzal Chaudhry, Resources, Data curation, Investigation, Methodology, Project administration, Writing - review and editing; John R Bradley, Conceptualization, Resources, Data curation, Investigation, Methodology, Writing - review and editing; Gregory J Hannon, Resources, Data curation, Supervision, Investigation, Methodology, Writing - review and editing; Gordon Dougan, Giles Wright, Conceptualization, Data curation, Supervision, Investigation, Methodology, Project administration, Writing - review and editing; Kenneth GC Smith, Conceptualization, Data curation, Investigation, Methodology, Project administration, Writing - review and editing; Paul J Lehner, Conceptualization, Supervision, Investigation, Methodology, Project administration, Writing - review and editing; Nicholas J Matheson, Data curation, Formal analysis, Investigation, Methodology, Project administration, Writing - review and editing; Stephen Baker, Conceptualization, Data curation, Formal analysis, Supervision, Funding acquisition, Investigation, Methodology, Writing - original draft, Project administration, Writing - review and editing; Michael P Weekes, Conceptualization, Data curation, Formal analysis, Supervision, Funding acquisition, Investigation, Visualization, Methodology, Writing - original draft, Project administration, Writing - review and editing

## Author ORCIDs

Nick K Jones (ID) https://orcid.org/0000-0003-4475-7761
Sushmita Sridhar (ID) http://orcid.org/0000-0001-7453-7482
Gregory J Hannon (ID) http://orcid.org/0000-0003-4021-3898
Ian G Goodfellow (ID) http://orcid.org/0000-0002-9483-510X
Paul J Lehner (ID) https://orcid.org/0000-0001-9383-1054
Nicholas J Matheson (ID) https://orcid.org/0000-0002-3318-1851
Michael P Weekes (ID) https://orcid.org/0000-0003-3196-5545

## Ethics

Human subjects: As a study of healthcare-associated infections, this investigation is exempt from requiring ethical approval under Section 251 of the NHS Act 2006 (see also the NHS Health Research Authority algorithm, available at http://www.hra-decisiontools.org.uk/research/, which concludes that no formal ethical approval is required). Our study was performed as a service evaluation of the Cambridge Universith Hospitals NHS Foundation Trust screening programme. The service provided was not changed in any way in order to undertake this evaluation.

Decision letter and Author response
Decision letter https://doi.org/10.7554/eLife.59391.sa1
Author response https://doi.org/10.7554/eLife.59391.sa2

---

## Additional files

### Supplementary files
• Transparent reporting form

### Data availability
All data generated or analysed during this study are included in the manuscript and supporting files.

---

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
