## [Decision Letter]

Thank you for submitting your article "Effective control of healthcare worker SARS-CoV-2 transmission in a period of declining community prevalence of COVID-19" for consideration by *eLife*. Your article has been reviewed by three peer reviewers, and the evaluation has been overseen by Jos van der Meer as the Senior and Reviewing Editor. The following individual involved in review of your submission has agreed to reveal their identity: Deenan Pillay (Reviewer #1).

The reviewers have discussed the reviews with one another and the Reviewing and Senior Editor drafted this decision to help you prepare a revised submission. Please note that there are serious concerns about the ethical aspects of the studies.

As the editors have judged that your manuscript is of interest, but as described below that additional data are required before it is published, we would like to draw your attention to changes in our revision policy that we have made in response to COVID-19 (https://elifesciences.org/articles/57162). First, because many researchers have temporarily lost access to the labs, we will give authors as much time as they need to submit revised manuscripts. We are also offering, if you choose, to post the manuscript to bioRxiv (if it is not already there) along with this decision letter and a formal designation that the manuscript is "in revision at *eLife*". Please let us know if you would like to pursue this option. (If your work is more suitable for medRxiv, you will need to post the preprint yourself, as the mechanisms for us to do so are still in development.)

Summary:

This is a nice addition to the earlier work published by the team demonstrating high rates of asymptomatic infection in HCW in one hospital. This Research Advance demonstrates an overall lower rate of infection, in conjunction with documented reductions across the UK. This is supportive of the hospital not acting as an independent hub for ongoing infection. The data are useful, but not unexpected.

Essential revisions:

1) There are a mixture of self-swabs and swabbing by a health worker. Has any comparison been made for known positives? Or an assessment of similar cell numbers?

2) Why combine symptomatic and contacts for analysis purposes?

3) We are concerned about those testing positive continuing to work – particularly since no ethics was obtained (exemption is cited)."Individuals with repeat positive screens that were not scheduled to come into close contact with heavily immunocompromised patients were advised to continue working, provided they remained asymptomatic". South Korea data are mentioned as a justification – but there does not seem to be logic to the recommendation used in this case. If immunocompromised patients are at risk, then all patients are at risk. What about those with specific and known risk factors for COVID-19 disease? The reviewers feel that ethical approval would be needed for a study like this.

Regarding these ethical issues we also asked the advice of an expert in medical ethics. The answer was: "I have tried to interpret the decision tool they refer to in their paper, which explains when something is research that is in need of formal ethics review. The authors could be right that their study is exempt from review, because it is service evaluation or usual practice in public health interventions. I am not an expert in British local requirements, so the authors are requested to explain explicitly why they did not obtain consent, and to explain which category of exemption they refer to.

I also agree with you that immunocompromised patients are not the only vulnerable group in hospitals, what about transmission to other categories of patients? Overall, the authors are very concise in their explanations and in giving context. It would be useful if they would reflect on these issues, and then I am happy to review their answer".

4) Even if ethics is not needed for HCW, then what about screening their household contacts, i.e., asymptomatic participants not associated with the hospital? What advice was given to those positive, and follow up?

5) A far smaller number of household contacts was tested than would be expected – why is that?

6) There is a need for the denominator of total staff and household contacts, rather than the positives/all tested.

7) Are there any further details that could be given on the HCW who were tested? E.g. what proportion of positives in the asymptomatic arm later became symptomatic? Was there any association between positivity and characteristics like use of PPE, role in the hospital (e.g. job description, whether they had direct contact with COVID patients etc.), age/gender?

8) "Effectively, the reproduction number (R) for SARS-CoV-2 amongst the

population of HCW in our hospital has been maintained <1". This statement feels oversimplified. Since there is a constant flow of patients from and to the community from hospitals, the reproduction number among HCW depends on (i) R between HCWs, (ii) R between HCWs and patients, (iii) R between hospitals and the community, (iv) R within the community. It is not clear which of these components of R the statement refers to, and whether the authors can even make such a statement given the data they have. For instance, positivity among HCWs may be driven by lower prevalence in the community or among patients, even while R between HCWs remains above 1.

9) Do clusters refer to sequencing? In which case need methods and data to show these clusters.

10) Subsection “Management of HCW with repeat positive tests”: On the issue of lack of evidence of onward transmission from repeat-positive asymptomatic individuals, this depends crucially on whether these individuals are detected positive after recovering from symptomatic illness, just before developing symptoms or asymptomatic throughout their time of positivity. In particular, there is good evidence that viral load (and risk of transmission) is highest just before a patient develops symptoms (He et al., 2020, among others), so a single positive test in an asymptomatic HCW does not imply that they are safe to continue working.

---

## [Author Response]

Essential revisions:1) There are a mixture of self-swabs and swabbing by a health worker. Has any comparison been made for known positives? Or an assessment of similar cell numbers?

Thank you for helping us to clarify this point. Self-swabbing was utilised in both screening arms, and no swabs were performed by second parties. We have amended the Materials and methods to ensure this is clear. “All individuals in each arm of the programme performed a self-swab at the back of the throat then the nasal cavity, followed by RNA extraction and amplification using real-time RT-PCR.”

2) Why combine symptomatic and contacts for analysis purposes?

Our analysis aimed to compare trends in incidence of SARS-CoV-2 infection in symptomatic versus asymptomatic individuals tested through our screening programme, in relation to trends in the hospital’s patient population and the wider regional community. In order to draw a clear distinction between these groups, we combined the HCW symptomatic and HCW symptomatic household contact screening arms to reflect all individuals with self-reported symptoms at the time of testing. One important reason to make this distinction is that lower levels of test positivity could be reasonably expected in the HCW asymptomatic screening arm, as we previously observed (Rivett et al., 2020). In addition, the number of individuals tested screened (and number of positive results) was considerably smaller in the HCW symptomatic household contact screening arm.

We have added “(reflecting all individuals with self-reported symptoms at the time of testing)” to the Results section, to make the reason for this distinction clearer to the reader.

3) We are concerned about those testing positive continuing to work – particularly since no ethics was obtained (exemption is cited)."Individuals with repeat positive screens that were not scheduled to come into close contact with heavily immunocompromised patients were advised to continue working, provided they remained asymptomatic". South Korea data are mentioned as a justification – but there does not seem to be logic to the recommendation used in this case. If immunocompromised patients are at risk, then all patients are at risk. What about those with specific and known risk factors for COVID-19 disease? The reviewers feel that ethical approval would be needed for a study like this.

Thank you for asking us to reflect on these areas of our hospital policy. This policy, based on UK national guidance, was devised by the Occupational Health, Infectious Diseases and Infection Control teams, and was *not* part of our study, which was performed as a service evaluation (see also below and reviewer point 4). Our study had no role in defining the policy, which dictated the approach to return to work for HCW.

The policy for heavily immunocompromised patients is based on UK national guidance for staff returning to work with stem cell transplant recipients, as cited in the manuscript (National Institute for Health and Care Excellence (NICE), 2020). This requires both a negative test and that the HCW is well for seven days prior to return to work.

For other patients, the policy includes completion of a minimum self-isolation period of seven days prior to return to work and that the HCW was also well for at least 48h. Again, it is based on UK national guidance, which hinges on time elapsed since symptom onset, rather than duration of detectable viral RNA after an initial positive result (https://www.gov.uk/government/publications/covid-19-stay-at-home-guidance/stay-at-home-guidance-for-households-with-possible-coronavirus-covid-19-infection).

The UK national guidance specifically refers to people with a positive SARS-CoV-2 test result (“This guidance is intended for: people with symptoms of coronavirus (COVID-19) infection, who have received a positive test result…”), stating: “If you have had symptoms of coronavirus (COVID-19), then you may end your self-isolation after 7 days and return to your normal routine if you do not have symptoms other than cough or loss of sense of smell/taste. If you still have a high temperature, keep self-isolating until your temperature returns to normal.”

Based on published evidence available at the time our SARS-CoV-2 testing occurred, we agreed with the UK national guidance that both immunocompetent and immunocompromised patients were unlikely to be at risk of infection from exposure to HCW with detectable SARS-CoV-2 RNA beyond day 8 of symptom onset, as long as these HCW had been well for at least 48h prior to return to work. There had not been reported examples of SARS-CoV-2 infectivity beyond day 8 of symptom onset in ex-vivo or human observational studies, as we referenced (UK Government, 2020; Korea Centers for Disease Control and Prevention (KCDC), 2020; Wölfel, Corman and Guggemos, 2020).

To clarify the above, we have re-worded the Results section (“Of these, 4/7 (57%) were tested to determine their suitability to return to work with heavily immunocompromised patients, as dictated by UK national guidance.”) and the Materials and methods section (“In accordance with UK national guidance, individuals with repeat positive screens following a minimum period of seven days self-isolation were advised to continue working if they were not scheduled to come into close contact with heavily immunocompromised patients, provided they remained asymptomatic.”).

Regarding these ethical issues we also asked the advice of an expert in medical ethics. The answer was: "I have tried to interpret the decision tool they refer to in their paper, which explains when something is research that is in need of formal ethics review. The authors could be right that their study is exempt from review, because it is service evaluation or usual practice in public health interventions. I am not an expert in British local requirements, so the authors are requested to explain explicitly why they did not obtain consent, and to explain which category of exemption they refer to.

Many thanks for helping to clarify our methods. In this paper we present the findings of a service evaluation of our health care worker screening programme. We did not change the service provided in any way in order to undertake this evaluation and the conclusion from the NHS Health Research Authority/Medical Research Council decision tool (http://www.hra-decisiontools.org.uk/research/) was “Your study would NOT be considered Research by the NHS”. As explained in our original, linked publication (Rivett et al., 2020), inclusion into the testing programme was voluntary, either arranged by the individual contacting the Occupational Health service (if symptomatic), or volunteering for testing offered to all individuals working in a given ward during the time of sampling (for the HCW asymptomatic screening arm).

We have modified the text as follows:

Materials and methods “Staff screening protocols

We previously described protocols for staff screening, sample collection, laboratory processing and results reporting in detail (Rivett et al., 2020). […] In the latter, HCWs could volunteer to take part in a rolling programme of testing for all patient-facing and non-patient-facing staff working in defined clinical areas thought to be at risk of SARS-CoV-2 transmission.”

Materials and methods “Ethics and consent:

As a study of healthcare-associated infections, this investigation is exempt from requiring ethical approval under Section 251 of the NHS Act 2006 (see also the NHS Health Research Authority algorithm, available at http://www.hra-decisiontools.org.uk/research/, which concludes that no formal ethical approval is required). Our study was performed as a service evaluation of the CUHNFT screening programme. The service provided was not changed in any way in order to undertake this evaluation.”

I also agree with you that immunocompromised patients are not the only vulnerable group in hospitals, what about transmission to other categories of patients? Overall, the authors are very concise in their explanations and in giving context. It would be useful if they would reflect on these issues, and then I am happy to review their answer".

Please see our response above. Our approach to permitting return to work after a positive test result was based on our hospital policy, which was devised by the Occupational Health, Infectious Diseases and Infection Control teams, informed by UK national guidance (https://www.gov.uk/government/publications/covid-19-stay-at-home-guidance/stay-at-home-guidance-for-households-with-possible-coronavirus-covid-19-infection). HCWs involved in the direct care of haematopoietic stem cell transplant recipients are the only group for which there is specific guidance recommending that a negative SARS-CoV-2 PCR test is required prior to returning to work (National Institute for Health and Care Excellence (NICE), 2020). As we describe, available literature up to the date of our study suggested that despite SARS-CoV-2 RNA remaining detectable for several weeks after symptom onset, individuals were unlikely to be infectious beyond day 8 (UK Government, 2020; Korea Centers for Disease Control and Prevention (KCDC), 2020; Wölfel, Corman and Guggemos, 2020).

4) Even if ethics is not needed for HCW, then what about screening their household contacts, i.e., asymptomatic participants not associated with the hospital? What advice was given to those positive, and follow up?

Many thanks for the comment. The same considerations as detailed in our reply to reviewer comment #3 apply. Household contacts were only tested if they were symptomatic and the HCW was consequently in self-quarantine because this contact had symptoms. Referral to the service was again voluntary, and offered via the hospital’s programme. Our study simply evaluated this service. There were no asymptomatic household contacts screened. This is explained in more detail in our original paper (Rivett et al., 2020). We have added further details explaining this into the ‘Staff screening protocols’ part of the Materials and methods section to aid clarity as a standalone paper:

“Two parallel streams of entry into the testing programme included (i) HCW symptomatic, and HCW symptomatic household contact screening arms and (ii) an HCW asymptomatic screening arm. […] In the latter, HCWs could volunteer to take part in a rolling programme of testing for all patient-facing and non-patient-facing staff working in defined clinical areas thought to be at risk of SARS-CoV-2 transmission.”

Advice to those testing positive was described in our original, linked manuscript (Rivett et al., 2020):

“Results reporting

As soon as they were available, positive results were telephoned to patients by Infectious Diseases physicians, who took further details of symptomatology including timing of onset, and gave clinical advice (Table 2). […] Verbal consent was gained for all results to be reported to the hospital’s infection control and health and safety teams, and to Public Health England, who received all positive and negative results as part of a daily reporting stream.”

5) A far smaller number of household contacts was tested than would be expected – why is that?

Testing of symptomatic household contacts (HHCs) was reliant on referral to the service by a co-habiting hospital employee. Uptake to this arm of the screening programme was lower than for self-referral by symptomatic HCWs, which may have either reflected a lack of awareness that symptomatic HHCs were eligible for testing (despite frequent advertising within the hospital), that non-hospital employees may have been more inclined to attend national testing centres, or that these individuals may have been less aware of all of the potential symptoms of COVID-19.

We have added lines to the Discussion section to highlight this point: “In addition to asymptomatic screening, testing of symptomatic HCWs is essential for preventing excessive erosion of the hospital workforce by self-isolation on the basis of symptoms alone, and testing of symptomatic HCW household contacts negates the need for unnecessary self-quarantine periods for co-habiting HCWs. […] Many non-hospital employees may also have been more inclined to attend national testing centres, or may have been less aware of all of the potential symptoms of COVID-19.”

6) There is a need for the denominator of total staff and household contacts, rather than the positives/all tested.

Many thanks for this point. CUH employs 11,000 members of staff, which we describe in the Discussion. The denominator for total staff household contacts is not recorded, as staff are not asked about how many others live in their household.

7) Are there any further details that could be given on the HCW who were tested? E.g. what proportion of positives in the asymptomatic arm later became symptomatic? Was there any association between positivity and characteristics like use of PPE, role in the hospital (e.g. job description, whether they had direct contact with COVID patients etc.), age/gender?

Many thanks for this suggestion. We have now included age and gender of those who tested positive in the Results section. We have also added a more detailed breakdown of the 21 positive tests from individuals in the *HCW asymptomatic screening arm*, which highlights that two were pre-symptomatic at the time of testing:

“In the *HCW asymptomatic screening arm,* 21/2,611 (0.8%) tests were positive, which again was significantly lower than 31/1,032 (3%) in the original study period (Fisher’s exact test p<0.0001). […] Our approach to patients with repeatedly positive SARS-CoV-2 PCR tests is described in the Materials and methods.”

We previously performed a detailed analysis of HCW working on ‘red’ vs. ‘green’ wards in our linked study (Rivett et al., 2020), and did not think it particularly informative to repeat this on the smaller sample of positive tests in the current analysis. Similarly, details of PPE used in all areas was previously discussed in detail. Of note, a minimum of masks, disposable aprons and gloves were worn in all patient-facing areas. However, the suggestion to add details of job description was interesting. We have added a table of the roles of all HCW testing positive in this and our previous linked study (Rivett et al., 2020), categorised according to screening arm (new Table 1):

“Table 1 summarises the total number of HCWs testing positive through either arm of the screening programme, according to job role. […] Reasonable comparison of the proportions testing positive through the HCW asymptomatic screening arm was not possible due to non-random sampling of different areas of the hospital, meaning some job roles had been more frequently targeted for asymptomatic screening than others.”

8) "Effectively, the reproduction number (R) for SARS-CoV-2 amongst thepopulation of HCW in our hospital has been maintained <1". This statement feels oversimplified. Since there is a constant flow of patients from and to the community from hospitals, the reproduction number among HCW depends on (i) R between HCWs, (ii) R between HCWs and patients, (iii) R between hospitals and the community, (iv) R within the community. It is not clear which of these components of R the statement refers to, and whether the authors can even make such a statement given the data they have. For instance, positivity among HCWs may be driven by lower prevalence in the community or among patients, even while R between HCWs remains above 1.

We agree that the original statement was simplified, and we agree in general terms with much of the reviewer’s helpful analysis.

In practice, it is common to refer to a reproduction number (R) for a population which is not completely isolated, such as a region within a country. For example, R in London is currently estimated to be 0.9. In this sort of usage, R is typically taken to refer to transmission between members of the identified population, and that is the way in which we use it here, i.e. in respect of transmission between members of our population of HCW.

Since the prevalence of infection in HCW is declining, we know that, on average, the number of secondary infections amongst HCW arising from each infected HCW *must* be <1. Put another way, the “R between HCW” (as defined by the reviewer) *must* be <1. If that wasn’t the case, the prevalence of infection in HCW would be increasing, rather than declining.

Again as stated by the reviewer, new infections amongst HCW may include instances of transmission from patients or other individuals outside the hospital, as well as transmission from other HCW. These introductions mean that the “R between HCW” is, in practice, almost certainly even lower that it appears.

We have adjusted the text in the Discussion to emphasise these points:

“Our data demonstrate a dramatic fall in the prevalence of symptomatic and asymptomatic SARS-CoV-2 infection amongst HCW in our hospital during the study period. […] For each of these clusters, timely identification of HCW infection proved effective in terminating chains of hospital transmission between staff, preventing ongoing nosocomial infection.”

9) Do clusters refer to sequencing? In which case need methods and data to show these clusters.

Apologies for omitting this definition. Clusters were defined pragmatically as three or more positive HCWs working in the same clinical area within a two week period, and did not rely on sequencing. We have added this definition into the Materials and methods section:

“Cluster investigation was initiated when three or more HCWs working in the same clinical area tested positive for SARS-CoV-2 within a one week period.”

10) Subsection “Management of HCW with repeat positive tests”: On the issue of lack of evidence of onward transmission from repeat-positive asymptomatic individuals, this depends crucially on whether these individuals are detected positive after recovering from symptomatic illness, just before developing symptoms or asymptomatic throughout their time of positivity. In particular, there is good evidence that viral load (and risk of transmission) is highest just before a patient develops symptoms (He et al., 2020, among others), so a single positive test in an asymptomatic HCW does not imply that they are safe to continue working.

Thank you for querying this. Repeat positive individuals were only advised to continue working if they had already completed a minimum of seven days in self-isolation, were asymptomatic upon completing their period of self-isolation, and were not scheduled to come into contact with heavily immunocompromised patients. We have re-worded the Materials and methods section to add clarity:

“In accordance with UK national guidance, individuals with repeat positive screens following a minimum period of seven days self-isolation were advised to continue working if they were not scheduled to come into close contact with heavily immunocompromised patients, provided they remained asymptomatic.”

Furthermore, all asymptomatic individuals testing positive for the first time were advised to self-isolate for a minimum of seven days from the time of sample collection, *or* from the onset of symptoms if they subsequently developed symptoms after the test. This approach was described in detail in our original, linked publication (Rivett et al., 2020, Table 2, ‘asymptomatic’ row). This approach was more cautionary than the national guidance, which stipulates a minimum of seven days isolation in total for individuals who test positive (https://www.gov.uk/government/publications/covid-19-stay-at-home-guidance/stay-at-home-guidance-for-households-with-possible-coronavirus-covid-19-infection).